# Marketability Discount in Various Economic Environments. Comparison of Developed and Emerging Markets on the Example of the USA and Poland

**Radosław Pastusiak \***, **Jakub Keller** and **Michał Radke**

Department of Corporate finance, University of Lodz, 90-412 Lodz, Poland; jakub.keller@uni.lodz.pl (J.K.); michal.radke@uni.lodz.pl (M.R.)

\* Correspondence: rpastusiak@uni.lodz.pl; Tel.: +48-42-635-48-32

**Abstract:** The aim of the presented article is to compare and evaluate the occurrence and level of marketability discount in developed and emerging markets in the example of the United States of America (USA) and Poland. According to the hypothesis put forward in the article, due to the smaller degree of development and depth of emerging markets, the marketability discount obtained in the context of the initial public offering (IPO) is lesser in its extent, as compared to the case when the IPO takes place in the developed market. The authors have made a statistic and econometric analysis based on a sample of nearly 200 IPOs in Poland and 1200 IPOs in the USA. The study used an analysis of the statistical differences between the groups (*t*-test), and also a linear modelling of the determinants of liquidity discount volume. The obtained results show that the stated hypothesis was correct, and that there are significant differences between the studied markets in reference to the marketability discount. The authors also concluded that the discount is not related to the condition of the company.

**Keywords:** company valuation; pricing; marketability discount; stock comparison; IPO

---

## 1. Introduction

This article presents the essence of the discount for lack of marketability. The basic difference between liquid and non-liquid shares is that the value of the former is greater. The ability to liquidate shares at a given time by a shareholder through the public market is a significant positive feature of a given company, and affects a higher price of the company compared, to analogous companies not listed in public markets. The aim of the article is to indicate the differences in the scope of discount for the lack of marketability between developed and emerging markets, and to provide the determinants of the size of this discount.

In the article, the authors put forward the following hypothesis on account of the smaller degree of development and maturity of emerging markets: the liquidity discount obtained in the context of initial public offering (IPO) is reduced to a lesser extent than if the IPO took place in the developed market.

Bonuses and discounts are used as adjustments to the underlying value of the company to determine the final price. In practice, they most often take a percentage form.

In the valuation of enterprises, the most commonly used discounts are (Trugman 2012):

- Lack of control discount;
- Discount for lack of marketability (illiquidity);
- Control premium;

- Key person discount;
- Private company discount;
- Discount from net asset value.

Marketability is defined by the International Glossary of Business Valuation Terms as the ability to quickly convert property into cash at minimal cost. The benchmark for marketability for business valuation is the market for active, publicly traded stocks. The holder can have the stock sold in less than a minute at or near the price of the last trade, and have cash in hand in three business days (Laro and Pratt 2005).

Discount for lack of marketability is defined by the International Glossary of Business Valuation Terms as an amount or percentage deducted from the value of an ownership interest to reflect the relative absence of marketability. The term 'relative' used in this definition usually refers to the value of the interest as if it was publicly traded, sometimes referred to as 'the publicly traded equivalent value' or 'the value if marketable' (Laro and Pratt 2005). Another definition created by Richard Sansing (1999) says that a discount for lack of marketability reflects the reduction in expected sales price, due to the lack of potential buyers. Damodaran (2005) states that liquid companies have a higher price than less liquid companies, so the discount for less liquid companies is higher.

Discount for lack of marketability means the difference in value of two similar equity securities, one of which represents a company listed on a public market, and the other non-listed company. As a result, due to the use of the discount for lack of marketability, you can make the pricing of shares in non-public enterprises on the basis of prices of shares of companies listed in public markets.

The scale of liquidity of a given asset depends primarily on the market's existence, such as a platform for exchanging commercial information and concluding transactions. For stock shares and enterprises, the market is the regulated securities trading systems, such as the Warsaw Stock Exchange (WSE). Other key factors affecting the level of liquidity are (Zarzecki 2014):

- Financial and economic condition (determining demand);
- Legal form of shares (ordinary shares, preference shares);
- Free float (number of shares in a distributed ownership structure).

In this chapter, the authors present what a 'discount for the lack of marketability' is and what affects its size. In the further part of the article, the authors present the most important research on the size of the discount; describe the methodology and results of their own research. In the next part, they open a discussion on the research on the discount and stimulate the conclusions on the research on the discount for the lack of marketability.

## 2. Literature Review

There is a lot of research in the literature about the discount for lack of marketability. Table 1 summarizes 20 restricted stock studies (i.e., 18 total studies, with two studies split into two subsets) that cover several hundred transactions spanning the late 1960s through 2013. These studies generally indicate a decrease in the average discounts for lack of marketability (DLOM) after 1990. The restricted stock transactions analysed in the studies covering the 1968 to 1988 period (where the average indicated DLOM was approximately 35%) were generally less marketable than the restricted stocks analysed after 1990 (where the average indicated DLOM was typically less than 25%) (Novak 2016).

**Table 1.** Restricted stock studies, summary of implied level of discounts for lack of marketability (DLOM).

| Restricted Stock Study | Observation Period of Study | Observed Average or Median Price Discount |
|---|---|---|
| SEC Overall Average | 1966–1969 | 25.8% |
| SEC Nonreporting OTC Companies | 1966–1969 | 32.6% |
| Milton Gelman | 1968–1970 | 33.0% |
| Robert R. Trout | 1968–1972 | 33.5% |
| Robert E. Moroney | 1969–1972 | 35.6% |
| J. Michael Maher | 1969–1973 | 35.4% |
| Standard Research Consultants | 1978–1982 | 45.0% |
| Willamette Management Associates | 1981–1984 | 31.2% |
| Hertzel and Smith | 1980–1987 | 20.1% |
| William L. Silber | 1981–1988 | 33.8% |
| Bajaj, Denis, Ferris, and Sarin | 1990–1995 | 22.2% |
| Johnson Study | 1991–1995 | 20.0% |
| Management Planning, Inc. | 1980–1996 | 27.0% |
| FMV Opinions, Inc. | 1980–1914 | 19.3% |
| Greene and Murray | 1980–1912 | 24.9% |
| Columbia Financial Advisors, Inc. | 1996–1997 | 21.0% |
| Columbia Financial Advisors, Inc. | 1997–1998 | 13.0% |
| LiquiStat | 2005–2006 | 32.8% |
| Angrist, Curtis, and Kerrigan | 1980–1909 | 15.9% |
| Stout Risius Ross | 2005–2010 | 10.9% |

Source: Novak (2016).

Except for research presented in the table the authors consider research of other authors. Koeplin et al. (2000), conducted a study for the discount of a private company. They examined 84 transactions in the USA, 108 foreign transactions, and the SDC database from 1984–1998. The authors stated that the Discount/Value method for the US market is on average 28.26%, while the median 30.62%, and for the foreign market is 43.87% and the median 5.96%. The discount calculated by the Value/Ebitda method for the US market is on average 20.39% and the median 18.14%, and for the foreign market it is 53.85%, and the median 23.49%. The discount calculated using the value/book value method for the US market is, on average, 17.81%, while the median is −7.00%, and for foreign markets it is 34.86% on average, while the median is 19.64%.

Chen and Xiong (2001) surveyed 2577 transactions from 18 brokerage houses in China in from October 2000 until July 2001, and they discounted the liquidity ratio between public transactions and private transactions (RIS) of the same companies. Their research shows that the average discount for RIS shares in relation to their public counterpart is 77.93% and 85.59%, accordingly, based on auctions and private transfers. Emory et al. (2002) carried out a survey of 543 transactions, of which 282 on shares and 261 on options in 1980–2000, in order to compare prices before IPO to IPO prices in a 5-month time horizon. Their research shows that the average discount for shares is 50%, and the median is 52%, while for the option the average is 43% and the median is 42%, while for the whole observation the average is 46%, and the median is 47%.

Pratt (2002), in his work, recalls research from 1980 to 2003, and states that the discount for lack of liquidity is from 30% to 50%, and the median is 40%.

Feldman (2004) states in his work that when shares changed the market from over-the-counter (OTC) to the New York Stock Exchange (NYSE), the price increased by 25%, which means a discount of 20%.

Officer (2007) stated that the discount for the liquidity of unlisted companies is, on average, 15% to 30%, compared to comparable public companies. He carried out his research in years 1979–2003 on the US market. He chose public companies and non-public companies (stand-alone private corporations and subsidiaries of other corporations) from the Securities Data Company (SDC) Mergers and Acquisitions database.

Reilly and Rutkowski (2007), in their text recall earlier research and state, on their basis, that until 1990, the discount was around 35%, and after 1990, around 25%. In the case of pre-IPO tests, the discount was 45–50%.

Inget (2009) has compared prices of private companies to the prices of public companies. The survey included 1125 companies from 2002–2007. His research shows that the median discount of a private company is −11.5%, that is, the company's private market price is 88.5% of the public company's price.

Harjoto and Paglia (2010) compared 431 matching pairs with Compusta and Pratt's 'Stats records information of Market Value of Invested Capital (MVIC) from 1994–2008', and observed the existence of a market value discount for private companies in relation to the market value of public companies, and so it was on average 75% with DLOMSALE[1], and median 68%, whereas with DLOMEBITDA[2] it was on average 50% and median 25%. From the regression model, they concluded that public buyers pay 14% to 15% less discount in comparison to private buyers, but the impact of public buyers is only statistically significant at 10%.

Torchio and Surana (2014) examined 1,000,000 observations from the American Stock Exchange (AMEX) and the National Association of Securities Dealers Automated Quotations (Nasdaq) market from 1926 to 2010. Their research involved measuring liquidity with the average monthly turnover of shares in each quarter. From their studies, the following liquidity-providing shares have a liquidity premium of less than 1%, the liquidity premium for low-liquidity shares is over 7%.

Mielcarz (2014) conducted a study in which he compared the prices of the first IPO listing with the price from the IPO prospectus, using the company's fair value the most advantageous market discount (MAMD) approach, in line with International Financial Reporting Standards (IFRS). His research included the IPO from the London Stock Exchange (LSE), the Frankfurt Stock Exchange (FSE), the NYSE Euronext Paris, the SWX Swiss Exchange and the Warsaw Stock Exchange in 2010–2013. Thus, the discount results for LSE by the MAMD method were 2.76%, while the most advantages market discount calculated based on weighted average (MAMDwa) method was 8.15%; for the FSE market, MAMD was 4.22%, MAMDwa was 6.31%; NYSE Euronext Paris was 3.34% discount with MAMD and MAMDwa is 8.18%; discount for the Swiss SIX by the MAMD method was 4.75% and 6.55% by the MAMDwa%; while for the WSE the discount method MAMD as 5.43% and the MAMDwa method was 6.39%.

Sosnowski and Wawryszuk-Misztal (2019), who examined the relationship between diversity on management and the quality of financial forecasts after IPO, also dealt with the problem of decomposing price changes after the IPO. Their research is one of the few attempts to assess this phenomenon in Poland. Although their research is not geared to the same goal as set out in this article, it sheds light on the problem of decomposition of the studied phenomenon. Sosnowski (2018) also notes that IPO companies are engaging in profit management to inflate the issue price. While such actions may benefit stock sellers at a public offering, they will in the long run bring about negative changes in the

---

[1]  DLOMSALE (%) = [1 − (MVIC/Sale for private firm)/(MVIC/Sale for public firm)] × 100
[2]  DLOMEBITDA (%) = [1 − (MVIC/EBITDA for private firm)/(MVIC/EBITDA for public firm)] × 100

company's results and goodwill. Borrowing future profits has its limitations, and reporting higher profits before IPOs causes their excessive decline in the next period.

Buchner (2016), in his work, examined the size of the discount for the lack of liquidity of private equity funds. His research showed that the upper limit of the discount for lack of liquidity is 7%. A similar discount was observed in venture and buyout funds.

Chen et al. (2017) studied the Korean banking market, and they managed to classify it into three categories of liquidity, and they claimed that the discount is highly correlated with the size of the bank and its ROE. Hur and Chung (2018) also studied the Korean market. They concluded that the mean liquidity premium is about 38% in their research sample.

Abudy et al. (2018), in their work, estimated that the discount for liquidity at 36.6%. In the same year, Wiśniewski (2018) made a study based on a global literature review, and supplemented this with his own research on the liquidity discount. He concluded that this type of discount is affecting the valuation in range of 9–11%.

Lopez and Martín (2019) analysed 824 public and private acquisitions on the Spanish market in 2006–2017, to determine the discount for marketability. Therefore, for industries such as agriculture, forestry and fisheries, you can get a 75% discount, moreover, if the buyer is a financial or professional institution, the primacy of marketability drops to −42%.

As can be deduced from the examples of research, the marketability discount or the lack of marketability exist, although it varies from several to even several dozen percent, depending on the survey, market or period.

## 3. Methodology

According to the proposed objective of the study, for which the authors want to identify the differences and determinants of the marketability discount on the market in Poland and the United States (US), the changes in share prices of companies that have decided to enter the public stock market were analysed. According to the literature listed above, the marketability discount is largely offset by the US market, which is considered to be the most mature of all available securities markets in the world. For this reason, the authors expect that the differences in share price changes for companies listed on the US market in the first few months after the IPO date should be objectively higher than would be recorded in the Polish market. The Warsaw Stock Exchange is in this case a typical representative of the stock exchange relatively well developed in the emerging market segment; however, due to significant differences in the size and activity of the markets in Poland and the USA, these exchanges are almost unmatched in a direct way. Although the company's public disclosure, by issuing securities on the stock exchange adequate to a given economy, will always be associated with an increase in prestige and a significant improvement in trading capacity, this is due to differences in the operation of individual securities exchanges, in particular in the number of participants and investors, the perception of this systemic change in the liquidity of trading in the company is significantly different for emerging and developed markets.

The study analysed the stock exchange debuts of companies from the American Stock Exchange and the Polish Stock Exchange in the last 10 years (2009–2019). In the case of the WSE, there were 193 debuts during the period under consideration. It should be noted here that their number in individual years is not equal, due to the changing phase of the economic situation and the condition of issuers. The largest number of debuts was recorded in 2011 (29) and the smallest in 2018 (6). The reference group to which the debuts on the Warsaw market were compared was the USA group of the 1206 IPO. It should also be noted that in the case of the selection of the reference group, only those debuting on the market in the USA are selected, which function in ten broadly defined economic areas, convergent with those that were represented in a sufficient numerous manners on the Polish market in the same period. The general areas defined by the authors are mentioned in Table 2.

**Table 2.** General economic sectors defined and included in the study.

| Economic Sectors | | | |
|---|---|---|---|
| IT | finance | media | industrial production |
| biotechnology | retail | medicine | food production |
| building construction | investments | Real estate | business services |
| power generation industry | logistics | clothing | mining |

Source: Own study.

The purpose of the division of the surveyed population into particular economic sectors is to indicate the differences in the value of debuting enterprises in respect to the differentiation into segments of activity.

The perception of a marketability discount assumes that the basic factor determining the change in the value of an unlisted entity in relation to a stock exchange listed is the fact of the possibility of buying and selling shares in this entity relatively easy. However, it is obvious that the debut time, market sentiment and the condition of the debuting company will have an impact on the change in share prices of individual companies. However, from the point of view of the company's analysis in fundamental terms, the condition of the company and its value usually do not change significantly from the perspective of several days to several months. This is related to the fact that the companies making their debut on the stock exchange are already well developed, and changes taking place in them require time (exceptions may be one-off, dynamic and unexpected economic and legal phenomena drastically changing the perspective of a given company). Therefore, it is assumed that in the first weeks of listing of a given entity after the IPO date, its economic situation does not change significantly, and that the differences in the value of its shares result mainly from current market trends and increased interest of investors in the new entity. The fact of this interest is also a marketability discount that affects non-listed companies.

Therefore, the authors analyse the price volatility of individual enterprises in the perspective of up to 125 trading sessions since the company's entry into the stock exchange. The analysed price changes are compared to the value of shares of individual entities that were determined in the process of company pricing during book building. The analysis period is set for 125 sessions, because of the fact that they consider the maximum that is not burdened with the appearance of new significant financial data that enterprises update under Polish conditions.

Another factor that the authors wanted to eliminate from the study of changes of individual companies is the fact that they belong to a specific sector, whose current changes may result in specific and involuntary changes in the price of a new listed entity. Hence, the study of price volatility also includes the rates of return from the main index of the Warsaw Stock Exchange (WIG) and sector-specific beta factors adequate for individual enterprises. The change of the share values of a given company, which we will identify as the marketability discount, can be recorded in the following way:

$$\text{marketibility discount} = \left( \frac{C_{in}}{C_{i0}} - 1 \right) - \left[ \beta_i * \left( \frac{C_{WIGn}}{C_{WIGipo}} - 1 \right) \right]$$

where,

$C_{in}$—price of *i* company in n stock sessions from IPO;

$C_{i0}$—price of *i* company before IPO;

$\beta_i$—factor beta adequate for *i* company;

$C_{WIGn}$—price of WIG index in n stock sessions from *i* company IPO;

$C_{WIGipo}$—price of WIG index at the day of the *i* company IPO.

The marketability discount, defined in this way, has been analysed in relation to the surveyed 193 entities. Then, the size of the discount obtained was compared in general terms—and in the particular

sectors—to the marketability discount for US companies. The discount size was obtained directly from the BVR Mergerstat "Valuation Advisors Lack of Marketability Discount Study". Observations of the US market were grouped according to the sector categories described earlier for the Polish market. The obtained marketability discount rates were compared with the use of parametric medium-difference tests between the surveyed Polish and American markets. The tests were performed both in general and in detailed terms, considering the breakdown into the sectors studied.

In the second part of the study, an analysis of factors potentially determining the amount of the marketability discount premium has been made. A simple econometric model of the least squares method was used for modelling. The previously defined discount set for the Polish market companies was treated as an explained variable. The explanatory variables were 38 parameters directly related to the financial situation of the debuting enterprise and the sector to which it belonged. Financial variables were obtained from Thomson Reuters databases. The financial parameters used in the study are presented in the Table 3.

**Table 3.** Financial explanatory variables used in the study.

| Financial Explanatory Variables | | | | | | |
|---|---|---|---|---|---|---|
| BV/Share | Total Capital | Common Equity | Total Assets | Revenue Per Share | Cash Flow | Current Ratio |
| EBIT/Tot Assets | Quick Ratio | Working Capital | ROA | Beta | Sharpe Ratio | Enterprise Value |

Source: Own study.

The parameters indicated above were used in modelling as explanatory variables both convergent with the IPO year, as well as delayed one year before IPO. In addition, the study used a set of binary variables referring to the 9 economic sectors among the companies included in the IPO database for the Polish market. The selection of the indicated industries is dictated by the availability of data regarding the liquidity discount for Polish companies. The authors used a maximum of a large number of observations (194), which allowed them to perform the research set out in the article. Due to the limited research sample, research conclusions will be drawn strictly for the sectors studied. The authors may hope that in the future, when the number of IPOs in Poland increases, it will be possible to cover more sectors with the analyses. Industries covered by the study are shown in Table 4.

**Table 4.** Sector included in the study as a binary variable.

| Sector | | |
|---|---|---|
| IT | investments | industrial production |
| finance | building construction | mining |
| medicine | real estate | food production |

Source: Own elaboration.

Using the above set of variables, linear modelling was performed, which in the understanding of the authors was to help in the assessment of the main determinants of the volume of the marketability discount.

## 4. Results

In accordance with the adopted analysis scheme, first of all, the comparison of all the observations concerning stock exchange debuts on the Polish and American market was made without dividing them into particular sectors. A summary of the basic characteristics in the surveyed population, along with the result of the test of significance of the difference between the average levels of marketability discount in these groups, is presented in the Table 5.

**Table 5.** Test of difference between mean marketability discount in Poland and USA.

| Null hypothesis: difference between two means = 0 |
| :---: |
| Sample 1: PL |
| n = 182, mean = 0.0649709, standard deviation = 0.722032 |
| Sample 2: USA |
| n = 1206, mean = 0.329001, standard deviation = 0.370105 |
| Test statistic: t(1386) = (0.0649709 − 0.329001)/0.0344037 = −7.67446 |
| Two-sided *p*-value = $3.116 \times 10^{-14}$ |

<div align="center">Source: Own study.</div>

According to data presented above, we conclude that the average level of the estimated discount under Polish conditions is around 6.5%, considering the factor modifying it for the change of the entire market. This means that, on average, the enterprises increased their prices abnormally over the first quarter by 6.5% in a situation where new financial data were not published from the moment of commencement to the end of measurement of price changes. It should be noted that the standard deviation in the studied group is very high, and amounts to approximately 72%. In the comparative group, referring to companies listed on the stock exchange in the United States, the level of the few month-long discount amounts to near 33% with a deviation of about 37%. This means that the discount in the American market is much higher and relatively more stable in relation to that in the Polish market. It should be noted that the difference in the discounted value studied between the Polish and American markets is very significant. This means that companies that enter the public market in the United States have a chance to get a higher premium for accessing to trade with their shares for the wide group of investors.

However, it should be considered whether the discussed phenomenon is homogeneous across the entire cross-section of the market, and whether it is possible to indicate the determinants of the volume of the studied discount. In the further part of the study, the authors compared the surveyed discount divided into selected industries.

They measured abnormal rates of return generated by the IPO for individual enterprises operating in various branches of the economy, and they carried out a check of the significance of differences between the levels of the Polish market discount versus the American market, split into sectors. Results are presented in Table 6.

On the basis of the above table, which presents the surplus of the rate of return identified with the marketability discount in terms of 1 m, 3 m and 6 m, it can be concluded that the generated surplus is relatively stable in the analysed period in general; however, the discount levels differ significantly between the industries, to a large extent.

The highest discount level in Polish conditions is recorded in the trade and mining industry. It is in these two market segments that enterprises have the opportunity to gain the most by public offering in the stock market. On the other hand, companies that perform significantly worse in the situation of transition to the public market belong to the media segment and the broadly understood investment industry (asset management). Further, the authors compared the marketability discount value in each of the specified industries between the Polish and American markets. The results are summarized in the Table 7.

**Table 6.** Levels of discounts divided by sector.

| Sector | 21 Trading Days Mean | 63 Trading Days Mean | 125 Trading Days Mean |
|---|---|---|---|
| biotechnology | 0.99% | 2.71% | −11.34% |
| building construction | 8.35% | 5.02% | −2.02% |
| power industry | 5.52% | 3.28% | 5.66% |
| finance | 2.13% | 0.11% | 1.39% |
| retail | 24.80% | 29.59% | 33.97% |
| investments | −4.74% | −7.98% | −24.97% |
| IT | 0.63% | −1.94% | −5.13% |
| logistics | −1.10% | 4.85% | 11.76% |
| media | −20.14% | −18.86% | −39.01% |
| medicine | 6.19% | 9.85% | 16.04% |
| real estate | 1.94% | −0.66% | −4.53% |
| clothing | 9.18% | −6.69% | −13.75% |
| industrial production | 2.51% | 9.59% | 39.18% |
| food production | 5.93% | 9.55% | 11.34% |
| business services | 6.53% | 4.41% | −1.66% |
| mining | 14.29% | 7.35% | −4.09% |
| average | 4.10% | 4.14% | 6.50% |

Source: Own study.

In the analysed group of 16 sectors, it was not possible to decide unequivocally about the differences in the marketability discount between the research and markets. The tests carried out, in seven cases, did not make it possible to determine the statistically significant difference between the given communities, although the nominal average in the subgroups seems to be significantly different, but, due to other features of the subgroup studied, the statistical significance of the differences could not be confirmed.

The authors believe that the lack of differences in some industries may be the result of the low sample size for the Polish market, which suggests careful inference about the discount in individual subgroups.

Finally, the authors focused on the issue of determinants of the surveyed discount in the perspective of up to 25 stock exchange sessions from IPO. Performed econometric estimations considering a number of factors, mainly of an internal nature, make it possible to draw interesting conclusions about legitimacy of perceiving above-average changes in rates of return in the context of marketability discount, and not seeing them in the specific condition and situation of a given enterprise. The initial model containing all variables included in the study is included in the Appendix A to this article. The Table 8 presents a final form obtained in the course of reduction of individual irrelevant variables, rejected in the course of subsequent estimations.

**Table 7.** Results of mean difference testing in each sector (H0: mean = 0)[3].

| Sector | Statistics | *p* Value |
|---|---|---|
| biotechnology | Sample 1: n = 18, mean = −0.0513263, stnd. dev. = 0.323332<br>Sample 2: n = 629, mean = 0.355227, stnd. dev. = 0.335888 | $p = 5.253 \times 10^{-7}$<br>Difference |
| Building construction | Sample 1: n = 5, mean = −0.13387, stnd. dev. = 0.229809<br>Sample 2: n = 59, mean = 0.354557, stnd. dev. = 0.47877 | *p* = 0.03526<br>Difference |
| energetics | Sample 1: n = 12, mean = −0.0202079, stnd. dev. = 0.127158<br>Sample 2: n = 12, mean = 0.389493, stnd. dev. = 0.352701 | *p* = 0.001016<br>Difference |
| finance | Sample 1: n = 7, mean = 0.0565698, stnd. dev. = 0.104122<br>Sample 2: n = 47, mean = 0.130446, stnd. dev. = 0.703936 | *p* = 0.7844<br>No difference |
| retail | Sample 1: n = 16, mean = 0.0138989, stnd. dev. = 0.266754<br>Sample 2: n = 139, mean = 0.244066, stnd. dev. = 0.262711 | *p* = 0.001149<br>Difference |
| investments | Sample 1: n = 7, mean = 0.339744, stnd. dev. = 0.491478<br>Sample 2: n = 8, mean = 0.355853, stnd. dev. = 0.232212 | *p* = 0.9351<br>No difference |
| IT | Sample 1: n = 9, mean = −0.249735, stnd. dev. = 0.396437<br>Sample 2: n = 25, mean = 0.136722, stnd. dev. = 0.277508 | *p* = 0.003168<br>Difference |
| logistics | Sample 1: n = 7, mean = 0.117627, stnd. dev. = 0.302287<br>Sample 2: n = 11, mean = 0.389082, stnd. dev. = 0.447672 | *p* = 0.1789<br>No difference |
| media | Sample 1: n = 3, mean = −0.390053, stnd. dev. = 0.254409<br>Sample 2: n = 33, mean = 0.396486, stnd. dev. = 0.261009 | $p = 1.694 \times 10^{-5}$<br>Difference |
| medicine | Sample 1: n = 13, mean = 0.160443, stnd. dev. = 0.33714<br>Sample 2: n = 110, mean = 0.32836, stnd. dev. = 0.34864 | *p* = 0.102<br>No difference |
| real estate | Sample 1: n = 22, mean = −0.0452965, stnd. dev. = 0.757989<br>Sample 2: n = 25, mean = 0.538579, stnd. dev. = 0.290108 | *p* = 0.000863<br>Difference |
| clothing | Sample 1: n = 5, mean = −0.137523, stnd. dev. = 0.237052<br>Sample 2: n = 11, mean = 0.363104, stnd. dev. = 0.192216 | *p* = 0.0004942<br>Difference |
| industrial production | Sample 1: n = 31, mean = 0.391798, stnd. dev. = 1.47392<br>Sample 2: n = 11, mean = 0.221475, stnd. dev. = 0.225539 | *p* = 0.7069<br>No difference |
| food production | Sample 1: n = 10, mean = 0.113371, stnd. dev. = 0.204949<br>Sample 2: n = 7, mean = 0.243991, stnd. dev. = 0.170824 | *p* = 0.1877<br>No difference |
| business services | Sample 1: n = 7, mean = −0.0166189, stnd. dev. = 0.32484<br>Sample 2: n = 53, mean = 0.392928, stnd. dev. = 0.272518 | *p* = 0.0005495<br>Difference |
| mining | Sample 1: n = 10, mean = −0.0408921, stnd. dev. = 0.359419<br>Sample 2: n = 26, mean = 0.211994, stnd. dev. = 0.755762 | *p* = 0.3204<br>No difference |

Source: Own elaboration.

Based on the final and reduced form of the model described above, the authors conclude that the majority of the parameters included in the study was not significant for shaping the marketability discount rate in Polish market conditions. It should be noted that the set of parameters referred, in a general way, to broadly understood comprehensive aspects of the company's operations: its size, liquidity, profitability and risk. We should conclude that the above-average rates of return equated with the marketability discount are independent of the financial aspects of the company in a broad sense, so they are obtained independently. In the understanding of the authors it is a premise that finding above-average rates of return is related to the fact of a change in the marketability of company's shares.

Referring to the final form of the econometric model, we can see that it indicates the statistical significance of the company's capital in both the IPO year and the previous year, as well as the

---

3    Sample 1, Normality Doornik-Hansen Test, *p*-value = 0.24812
    Sample 2, Normality Doornik-Hansen Test, *p*-value = 0.09189.

significant relationship between the generated surplus and affiliation to the investment sector or industrial production. It should be noted, however, that the coefficient of determination of the above model is about 7%, which is very low. In addition, the model described above is highly biased and even inconclusive, due to the occurrence of heteroskedasticity and the lack of normal distribution of model residues, which makes the estimated coefficients not interpretable.

**Table 8.** Model of the determinants of marketability discount in Poland.

| Variable | Coefficient | Standard Dev. | *t*-Test | *p*-Value |
|---|---|---|---|---|
| const | −0.000846322 | 0.0413 | −0.0205 | 0.98370 |
| Total Capital | $2.00442 \times 10^{-11}$ | $5.30 \times 10^{-12}$ | 3.7766 | 0.00022 |
| Total Capital (−1) | $-2.22686 \times 10^{-11}$ | $4.10 \times 10^{-12}$ | −5.4311 | <0.00001 |
| Investments sector (0/1) | −0.35513 | 0.116974 | −3.0360 | 0.00278 |
| Industrial production (0/1) | 0.500286 | 0.298777 | 1.6744 | 0.09589 |
| The arithmetic mean of the dependent variable | 0.059645 | standard deviation of a dependent variable | | 0.754125 |
| R-square | 0.072300 | Adjusted R-square | | 0.050342 |
| Logarithm of credibility | −190.7626 | Akaike criteria | | 391.5251 |
| Schwarz criteria | 407.3204 | Hannan-Quinn criteria | | 397.9327 |

Source: Own elaboration.

All the above elements induce the authors to state that the surpluses obtained after the IPO are objectively unrelated to the parameters of enterprises published on the stock exchange, and in accordance with the literature discussed earlier, may be related to a change in the trading capacity of the entities in question, i.e., in the scope of the discussed marketability discount.

## 5. Discussion

The proposed study addresses the subject of valuation in relation to the discount that accompanies non-public companies, i.e., those that have limited opportunities to trade their shares in relation to public companies. This issue is important both from the point of view of constructing predictions as to the change in the share price p assumed in the time perspective from the IPO, but also for the needs of the pricing companies, whose value of shares should be estimated for over-the-counter transactions. The authors see a strong need to conduct this type of research due to the lack of data on such an approach to the valuation of shares of OTC companies when their shares are sold to an undefined buyer in the transaction.

The analyses undertaken are an attempt to determine possible, statistically significant parameters that will allow the discount level for a private company to be determined, however, based on the practice observed for companies after IPO. The authors are aware that the subject matter is complex and requires the study of a large number of company parameters that may affect price changes, and whose recognition is possible in a synthetic and model way. Such an analysis was made in the presented article. However, it showed that a large part of the parameters tested did not constitute the basis for an unequivocal determination of the level of liquidity discount.

However, the authors do not consider the actions taken to be erroneous, because these variables were deliberately examined due to the researchers' objectives. However, the study showed that the adopted set of variables describes, to a small extent, the differences between the levels of liquidity discount. In the authors' opinion, this is also an important knowledge, because it indicates parameters that—in the presented approach—are not, in most cases, good indicators of the discount amount. This, in turn, facilitates further research and selection of variables for verification.

It is also worth referring to the issue of comparing American and Polish markets in the study. As indicated in the description of the survey method, the indicated markets are a fairly general representation of two types of exchanges and economies. The American market is the most liquid stock market in the world, which is why the research on debuts in it indicates the maximum level of profit that companies can achieve, thanks to the decision to make it public on the stock exchange in this country.

On the other hand, we are examining the Polish market, which is the strongest market in Central and Eastern Europe, which shows the perspective and possibilities of price bonuses due to debut in this segment of stock exchanges.

For this reason, according to the authors, it is interesting to compare these exchanges, because, apart from indicating the average levels of liquidity discount in the industries studied, we are able to assess whether in a given sector and in terms of marketability discount, emerging and developed markets are similar or different.

In this respect, the study made it possible to determine the industries in which debuts on the emerging market can give positive price effects similar to the developed market. Such industries are, for example, finances and investments. According to the authors, such a conclusion is a result of the specificity of the indicated industries. The financial and investment sector is so developed and computerized that development and global operations are similar, regardless of the main place of business. Hence, prospects of price changes after IPO may be similar for such companies in both developed and developing markets, despite significant differences in the volume of trading in exchanges.

On the other hand, we also have industries in which making the company public in the emerging market may give less positive price effects than if the IPO was in a developed market. Examples of these industries are the energy sector, media or mining. The authors believe that the convergence in these areas is primarily due to the nature of the industries themselves, their technological and regional level, as perceived by the community and investors. Significant discrepancies for the level of liquidity discount in these industries mean that when prices are set in OTC transactions, it should also be considered that the market on which the transaction is made may significantly affect the offer price.

The further part of the research also showed that of the examined variables determining the level of the discount level, the capital levels and belonging to two of the analysed sectors are significant. This means that previously determined differences between the surveyed industries must be further analysed in terms of the characteristics of selected companies participating in the study. These are certainly challenges for further research on the topic presented.

## 6. Conclusions

The study was primarily aimed at showing the appropriateness of considering the marketability discount in the case of enterprises that are not listed on the stock markets. In the study, the authors also wanted to show that an important element of the perception of the marketability discount is the region of the world in which the valued enterprise functions. Analysing the differences between the Polish market and the US market, they pointed to significant discrepancies in the market valuation of a company operating in the developed market and the emerging market. This is an important information, both from the point of view of potential investors and from the point of view of entities that are considering being public on the stock market. An important aspect of the considerations under the study is the fact, then they indicate the independence of the occurrence of the discount in question in relation to the condition of the examined enterprise. At the same time, this does not mean that authors are of the opinion that the company's condition has no impact on the rate of return it achieves in the long horizon after the IPO. The conducted research, however, made it possible to obtain a confirmation of the hypothesis that there are significant differences between the analysed markets, and that horizontal marketability discounts are significantly different in them, and this significantly influences the pricing process of the company, which operates under different conditions.

It should be remembered that the analysed variables were used to model the excess return over the index changes related to a specific industry, so, in the analyses undertaken, the authors do not refer to the total return rate achieved by a given company, only to the part that has connections with the fact of being public on the stock exchange. Hence the assumptions regarding the rate of return and the horizon of the study. The analyses described in the text indicated that surplus modelling is independent of variables related to the condition of a particular company.

In addition, the analyses are an important contribution to research in the pricing of enterprises that are not listed in public markets. The analyses carried out prove that in case of the evaluating of this type of entities it is necessary to consider the marketability discount, as the literature quoted above proves. However, the study carried out indicates, in a more precise manner, what the potential marketability discount rate should be considered in the case of such a company pricing, which belongs to a specific industry covered by the study conducted. The analyses presented here are in line with the trend of improving the quality of the evaluations of enterprises operating in the area of the emerging Eastern European markets.

At the same time, the authors are aware that the conducted analyses should be further deepened, especially by expanding the research sample both within the Polish capital market and considering other exchanges. It is also necessary to explore further the variables affecting the size of the marketability discount. The authors are aware that further search for parameters should, to a greater extent, also consider the issues related to trading parameters of individual entities listed on the markets, such as sales volume, number of investors involved, frequency of transactions and the size of packages that are exchanged.

The study revealed several interesting similarities and discrepancies between the Polish and US markets in the examined context of liquidity discount. However, the authors see significant limitations of the analysis carried out, which can be resolved in subsequent phases of research on the studied topic. First of all, it is necessary to search further for research observations on the Polish market, because in this study only 194 observations could be included, which is a significantly lower number than for the American market.

This problem is the effect of the decreasing attractiveness of the Polish stock exchange, which translates into a small number of new companies' debuts. In recent years, we have seen a number of IPOs between 10 and 30 per year. In addition, most of them concern the same industries—the medical sector and high technologies—which means that the sample is not diversified by industry, and it is very difficult to analyse the liquidity discount in the so-called traditional industries.

Certainly, the limitation of the study is also a predefined set of variables, according to which an attempt was made to decompose the examined discount. Future research should extend the catalogue of variables studied in order to find a more effective model determining the amount of marketability discount. The authors also assume methodological modifications involving the inclusion of ANOVA or CART trees in the study.

**Author Contributions:** Conceptualization, R.P., J.K. and M.R.; methodology, R.P., J.K. and M.R.; software, R.P., J.K. and M.R.; validation, R.P., J.K. and M.R..; formal analysis, R.P., J.K. and M.R.; investigation, R.P., J.K. and M.R.; resources, R.P., J.K. and M.R.; data curation, R.P., J.K. and M.R.; writing—original draft preparation, R.P., J.K. and M.R.; writing—review and editing, R.P., J.K. and M.R.; visualization, R.P., J.K. and M.R.; supervision, R.P., J.K. and M.R.; project administration, R.P., J.K. and M.R.; funding acquisition, R.P., J.K. and M.R. All authors have read and agreed to the published version of the manuscript.

**Funding:** This research received no external funding.

**Conflicts of Interest:** The authors declare no conflict of interest.

## Appendix A

**Table A1.** Initial model of the marketability discount.

| Variable | Coefficient | Standard Dev. | *t*-Test | *p*-Value |
|---|---|---|---|---|
| const | −9.079 | 95.832 | −0.095 | 0.925 |
| Year of IPO | 0.005 | 0.048 | 0.095 | 0.924 |
| Volume of public offer | 0.000 | 0.000 | −0.170 | 0.866 |
| BV per share | −0.042 | 0.027 | −1.573 | 0.122 |
| Total Capital | 0.000 | 0.000 | 0.263 | 0.794 |
| Common Equity | 0.000 | 0.000 | 0.783 | 0.438 |
| Total Assets | 0.000 | 0.000 | 1.627 | 0.111 |
| Revenue Per Share | 0.001 | 0.004 | 0.290 | 0.773 |
| Cash Flow | 0.000 | 0.000 | −0.154 | 0.878 |
| Current Ratio | −0.286 | 0.188 | −1.518 | 0.136 |
| EBIT To Assets | 1.066 | 1.167 | 0.914 | 0.366 |
| Quick Ratio | 0.306 | 0.210 | 1.459 | 0.151 |
| Working Capital | 0.000 | 0.000 | −1.500 | 0.140 |
| ROA | −0.010 | 0.012 | −0.855 | 0.397 |
| Beta | 0.145 | 0.215 | 0.675 | 0.503 |
| Sharpe Ratio | 0.671 | 1.021 | 0.657 | 0.515 |
| Enterprise Value | 0.000 | 0.000 | −0.269 | 0.789 |
| BV per share (−1) | 0.057 | 0.035 | 1.658 | 0.104 |
| Total Capital (−1) | 0.000 | 0.000 | −0.614 | 0.542 |
| Common Equity (−1) | 0.000 | 0.000 | −0.874 | 0.387 |
| Total Assets (−1) | 0.000 | 0.000 | −1.666 | 0.103 |
| Revenue Per Share (−1) | −0.004 | 0.007 | −0.666 | 0.509 |
| Cash Flow (−1) | 0.000 | 0.000 | −0.827 | 0.413 |
| Current Ratio (−1) | 0.124 | 0.236 | 0.525 | 0.602 |
| EBIT To Assets (−1) | −0.554 | 1.351 | −0.410 | 0.684 |
| Quick Ratio (−1) | −0.171 | 0.257 | −0.667 | 0.508 |
| Working Capital (−1) | 0.000 | 0.000 | 0.999 | 0.323 |
| ROA (−1) | −0.001 | 0.015 | −0.036 | 0.971 |
| IT | 0.091 | 0.641 | 0.142 | 0.888 |
| Investments | 11.682 | 15.451 | 0.756 | 0.453 |
| Industrial production | 0.092 | 0.218 | 0.421 | 0.676 |
| Real estate | 0.684 | 0.338 | 2.027 | 0.049 |
| Medicine | 0.042 | 0.322 | 0.129 | 0.898 |
| Finance | 0.420 | 0.498 | 0.843 | 0.404 |
| Building construction | −0.189 | 0.250 | −0.757 | 0.453 |
| Mining | −0.020 | 0.440 | −0.045 | 0.965 |
| Food production | 0.020 | 0.316 | 0.063 | 0.950 |

Source: Own elaboration.

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
