# Peer review of "Marketability Discount in Various Economic Environments. Comparison of Developed and Emerging Markets on the Example of the USA and Poland"

_jrfm, doi:10.3390/jrfm13060132_

Round 1

Reviewer 1 Report

Referee Report of “Marketability Discount in Various Economic Environments. Comparison of Developed and Emerging Markets on the Example of the USA and Poland.”

Objectives: This article estimates a linear model with various independent variables, including marketability discount on Poland and USA data to illustrate that access to liquidity is one of the factors that causes the discrepancies in the market valuation of a company operating on the developed market and the emerging market.

Comments:

  • In Table 2 you list the economic sectors in this study, and in Table 4, you assign the 9 most-representative economic sectors in Poland as dummy variables. How representative are these 9 sectors to the Poland economy? 70% of the entire dataset? 80%? Would the results change significantly if I include all the sectors as binary variables?
  • Different sectors have different results. For example, in Table 7 we do not observe any statistical difference in sectors such as investments, logistics, or food production. While other sectors, for instance, biotechnology sectors, we do observe the differences. What are the economic intuition behind these differences? Are these differences driven the main results on aggregate data?

Author Response

Dear Reviewer,

Thank you for your kind comments and valuable feedback on our paper. We worked through all suggestions thoroughly and answered each of your requests accordingly. In the following, we list in more detail our changes referring to your comments and suggestions.

  1. Literature review has been supplemented and expanded
  2. The method of selecting sectors for the study was explained in more detail.
  3. Discrepancies between the results of the analysis of individual sectors were explained and embedded in the context of the purpose of the study.
  4. A new subsection "discussion" has been added to the article, in which the authors explain the context and sense of the results more widely.
  5. The previously described acronyms used in the text have been explained.
  6. The bibliography has been verified and corrected.
  7. The summary of the article describes in more detail the limitations of the study and the possibilities of its future development.
  8. The methods used in article were described in abstract.
  9. The introduction now includes a description of the article structure.
  10. The description of the purpose of the work, which was mistakenly included twice in the introduction, has been corrected.
  11. The study took into account the result of the normal distribution analysis of the tested samples.
  12. Another language correction of the text has been made.

Reviewer 2 Report

The paper presents an interesting topic of “the essence of the discount for lack of marketability”, using the case of Poland and US markets. However, the overall quality of the paper is medium, it is based on classic methodology and the results are poor and not enough contextualized. The article does not provide original and new contribution to its theme.

Despite the promising implications of the paper and an overall sound methodological framework I rise a couple of points that might bring improvements to the paper

1. The authors find significant discrepancies in the market valuation of a company operating on the developed market (US) and the emerging market (Poland). The authors mentioned “this is important information both from the point of view of potential investors and from the point of view of entities that are considering being public on the stock market”. However, it is not clear which driving forces lead the effects of these differences. Moreover, the reader of the article learns absolutely nothing about the elements which constitute the particularity of these two markets and which could explain the observed differences. In addition, extending the case of Poland to all emerging markets and that of the United States to all developed countries is simplistic, and turns out to be a crude analytical approach.

  1. Many acronyms are used in the text without being explained.
  2. Is necessary to verify the references because some are not described according to the instructions for authors.
  3. The limits of the paper are not very clearly described.

Author Response

(The authors gave the same response as above.)

Reviewer 3 Report

Dear authors,

I have some recommendations for you:

  1. Abstract:
    Define methods used.

  2. Introduction:
    You have in this part the aim set twice, why?
    The aim of the study is to determine the discount for lack of marketability in emerging markets on the example of companies listed on the Warsaw Stock Exchange.
    The aim of the article is to indicate differences in the scope of discount for liquidity between developed and emerging markets and indication of the determinants of the size of this discount.
    Describe clearly the structure of the paper in this section and significance of your research.

  3. Literature review:
    19 used resources is not enough for this type of journal, please extend your review. I suggest
    Sosnowski, Tomasz, and Anna Wawryszuk-Misztal. 2019. Diversity on management and supervisory board and accuracy of management earnings forecasts in IPO prospectuses. Ekonomia i Prawo. Economics and Law 18: 347-363.
    Sosnowski, Tomasz. (2018). Earnings management in the private equity divestment process on Warsaw Stock Exchange. Equilibrium. Quarterly Journal of Economics and Economic Policy 13: 689-705.
    Kliestik, T., Misankova, M.,  Valaskova, K. & Svabova, L. (2018). Bankruptcy prevention: new effort to reflect on legal and social changes. Science and Engineering Ethics, vol. 24, No 2, pp. 791-803.
    Kovacova, M., Kliestik, T., Valaskova, K., Durana, P., & Juhaszova, Z. (2019). Systematic review of variables applied in bankruptcy prediction models of Visegrad group countries. Oeconomia Copernicana, 10(4), 743-772.
  4. Empirical analysis:
    Please, rename to Results.
    You have used t-test. Have you also tested normality, that is necessary assumption to run t-test?

  5. Discussion:
    None, without this crucial part is not possible to publish your paper.

  6. Conclusions:
    Highlight precisely limitations.

After reworking, I think the paper has its place in scientific journal JRFM.

I hope my comment will be useful for your future work.

Author Response

(The authors gave the same response as above.)

Round 2

Reviewer 2 Report

Changes made to the article by the authors are significant. These changes respond to most of the comments made previously.

Methodology and results are now adequately described. The new discussion section is short but provides some interpretation of results and findings.

Author Response

Thank you for your comments.

Reviewer 3 Report

Dear Authors,

Dear authors,

I appreciate your effort to improve your manuscript.

You have changed the manuscript aiming to include all of my comments and suggestions.

I recommend publishing.

Good luck in your future work.

Author Response

Thank you for your comments.